# Simulating the Self-Assembly and Hysteresis Loops of Ferromagnetic Nanoparticles with Sticking of Ligands

**DOI:** 10.3390/nano11112870

**Published:** 2021-10-27

**Authors:** Nicholas R. Anderson, Jonathon Davidson, Dana R. Louie, David Serantes, Karen L. Livesey

**Affiliations:** 1UCCS Biofrontiers Center, University of Colorado at Colorado Springs, 1420 Austin Bluffs Parkway, Colorado Springs, CO 80918, USA; nanders3@uccs.edu (N.R.A.); jdavids7@uccs.edu (J.D.); dana.r.louie@nasa.gov (D.R.L.); david.serantes@usc.es (D.S.); 2Instituto de Investigacións Tecnolóxicas and Applied Physics Department, Universidade de Santiago de Compostela, 15782 Santiago de Compostela, Spain; 3Department of Physics, University of York, York YO10 5DD, UK; 4School of Information and Physical Sciences, University of Newcastle, Callaghan, NSW 2308, Australia

**Keywords:** magnetic nanoparticle, Langevin simulation, ligand, magnetic hyperthermia

## Abstract

The agglomeration of ferromagnetic nanoparticles in a fluid is studied using nanoparticle-level Langevin dynamics simulations. The simulations have interdigitation and bridging between ligand coatings included using a computationally-cheap, phenomenological sticking parameter *c*. The interactions between ligand coatings are shown in this preliminary study to be important in determining the shapes of agglomerates that form. A critical size for the sticking parameter is estimated analytically and via the simulations and indicates where particle agglomerates transition from well-ordered (*c* is small) to disordered (*c* is large) shapes. Results are also presented for the hysteresis loops (magnetization versus applied field) for these particle systems in an oscillating magnetic field appropriate for hyperthermia applications. The results show that the clumping of particles has a significant effect on their macroscopic properties, with important consequences on applications. In particular, the work done by an oscillating field on the system has a nonmonotonic dependence on *c*.

## 1. Introduction

The interdigitation and bridging of ligand polymers has been shown to greatly influence how nanoparticles aggregate to form clumps, chains or even nanocrystals [1,2,3,4,5,6,7]. This is especially important in biological environments where nanoparticles develop a protein corona, resulting in a different aggregation that in turn affects how they perform diagnostic and therapeutic functions [8]. Here, we consider magnetic nanoparticles that have unique applications in fluids [9] such as in magnetic hyperthermia to treat cancer [10], guided drug delivery [11], lab-on-a-chip devices [12], and contrast agents [13].

It is well-known that when magnetic particles agglomerate, they behave very differently compared to isolated particles. For example, their effective magnetic anisotropy is altered [14] and their coercive fields and remanance magnetization (when subject to an alternating field) can be increased or decreased depending on the clump shape [15,16,17]. In order to predict and design materials with these properties, one must be able to correctly predict what types of aggregates (chains, nanocrystals, clumps, rings) do in fact form when ligand environments are altered. Until recently [3], very few simulations have included the role of ligand bridging or interdigitation. Without these effects, long chains of particles typically form in simulations of ferromagnetic particles with the magnetic dipole moments joining tip-to-tail. These chains can come together to form thicker bundles when the number of particles in an agglomerate becomes very large and it therefore becomes entropically favorable to do so compared to growing longer and longer single-width chains [18]. (This has been studied in more detail for superparamagnetic beads in an applied field [19], for which the physics are somewhat different.) We show here that ligand sticking can stabilize configurations of nanoparticles whereby the dipolar interactions are not minimized (i.e., magnetic dipoles are not aligned tip-to-tail) and so more disordered particle agglomerate shapes form.

We present particle-level Langevin simulations of ferromagnetic nanoparticles in a finite-temperature viscous fluid, interacting via magnetic dipolar fields, steric repulsion, van der Waals interactions, and ligand sticking (due to bridging and interdigitation). We treat the sticking using a phenomenological sticking parameter that essentially damps the motion of particles that are part of an agglomerate. In other words, when particles come into contact with each other, there is resistance to their individual rotation and translation due to ligands binding or simply becoming entangled with each other. The phenomenological sticking parameter represents a weighting factor in the Langevin equation, which is in contrast to previous methods to treat ligand sticking through an effective potential [3] or to consider that particles are rigidly locked together once they touch [20,21]. Our method has the advantage of being computationally cheap to implement, and it is more realistic than rigid-locking. We present preliminary results for 50,050 nm-diameter magnetite particles simulated for 0.01 s, with periodic boundary conditions. We also present an analytic estimate for the critical phenomenological sticking parameter that separates the formation of chains (low stickiness) from the formation of more disordered clumps (high stickiness). The analytic estimate agrees well with the simulation results and corresponds to realistic force magnitudes.

Results are also presented for when the particle system is subjected to an oscillating magnetic field with appropriate amplitude and frequency for magnetic hyperthermia treatment (100 kA/m at 1.25 kHz). The net magnetization as a function of the applied magnetic field is found and the resulting hysteresis loops show a marked difference when there is ligand friction compared to when there is not. For one thing, the area enclosed by the hysteresis loops—which is correlated with the heat that particles may generate—is increased and then decreased as the ligands are made more sticky.

## 2. Materials and Methods

### 2.1. Simulations

The translational equation of motion for a particle *i* is given by Newton’s second law in the Langevin form [22], namely
(1)$$\begin{array}{ccc} m\frac{d^{2}{\vec{x}}_{i}}{dt^{2}} & = & {\vec{F}}_{i}\left[{\vec{x}}_{j}\left(t\right)\right]-m_{i}\gamma_{t}{\vec{v}}_{i}\left(t\right)+{\vec{\zeta }}_{i}\left(t\right)\;\;\equiv {\vec{f}}_{i}, \end{array}$$
where *m* is the mass of all particles, ${\vec{x}}_{i}$ is the position of the particle, $F_{i}$ is the deterministic force on the particle due to the interactions with all the *j* other particles (detailed in the Appendix B), $\gamma_{t}$ is the translational viscous coefficient, ${\vec{v}}_{i}$ is the velocity, and ${\vec{\zeta }}_{i}$ is a random force that is connected to the viscosity via the fluctuation–dissipation theorem. $\vec{\zeta }$ is assumed to obey a Gaussian distribution with
(2)$$\begin{array}{ccc} \langle \zeta_{\alpha }\left(t\right)\rangle & = & 0,\;\;\;\;\;\;\langle \zeta_{\alpha }\left(t\right)\zeta_{\beta }\left(t+t^{\prime }\right)\rangle =2m\gamma_{t}k_{B}T\;\delta_{\alpha \beta }\;\delta \left(t^{\prime }\right), \end{array}$$
where $k_{B}$ is Boltzmann’s constant, *T* is temperature, and α and β represent components of the force. The drag coefficient is given by Stoke’s formula [22]
(3)$$m\gamma_{t}=6\pi \eta R,$$
where η is the viscosity of the fluid and *R* is the hydrodynamic radius of the particle, which is on the order of the particle radius *r* plus the ligand coating length *L*. In fact, hydrodynamic interactions between particles are present [23,24] (the disturbance of a particle moving through a fluid influences the motion of others), but here they are ignored as a first approximation and to isolate the effect of the ligand interactions.

We discuss the manipulation of Equation (Equation 1) to include phenomenologically the effects of ligand sticking. Note that the sum of all forces (deterministic, viscous and random thermal fluctuations) on the right-hand-side of Equation (Equation 1) have been denoted ${\vec{f}}_{i}$ for convenience.

If a particle *i* was rigidly stuck inside an agglomerate containing $N_{agg}$ particles, then it would have to move in exactly the same way as its neighbors and its equation of motion becomes
(4)$$mN_{agg}\frac{d^{2}{\vec{x}}_{i}}{dt^{2}}=\sum_{k=1}^{N_{agg}}{\vec{f}}_{k}.$$

This situation has indeed been considered by some authors and can be said to “coarse-grain” the system and speed up simulations [20,21,25]. However, it is more realistic to have particles inside agglomerates that are both influenced by their neighbors, but also have the ability to move independently because any ligand sticking force is not infinite in strength. In fact, we see in simulations that agglomerates change their shape over time, and do so depending on the strength with which their ligand coatings interact. For example, studies have found binding forces ranging from 100 to 700 pN over distances from 0 to 7 nm for PEG ligands attached to a surface [26,27]. One can take this as an estimate for the forces that may bind ligands to each other. These forces are on the order of the magnetic and thermal forces acting on particles (see Appendix B and parameters given at the end of this section).

To better model these finite ligand sticking strengths, we write a weighted average of Equations (Equation 1) and (Equation 4), namely
(5)$$m\frac{d^{2}{\vec{x}}_{i}}{dt^{2}}=\left(1-c\right){\vec{f}}_{i}+\frac{c}{N_{agg}}\sum_{k=1}^{N_{agg}}{\vec{f}}_{k},$$
where *c* is the weighting factor, which can range from zero (no sticking of particles) to one (infinite force sticking particles together when they come in contact). We call this parameter *c* the phenomenological sticking parameter. If two particles have their magnetic surfaces separated by distances less than 2L, where *L* is the ligand coating length, they are deemed to be part of the same agglomerate. One can check that Equation (Equation 5) preserves the laws of physics; for example, conservation of momentum is preserved.

When magnetic torques are appreciable, then the rotational equation of motion for particle *i* must also be considered [28]
(6)$$\begin{array}{ccc} I\frac{d{\vec{\omega }}_{i}}{dt} & = & {\vec{T}}_{i}\left[{\vec{x}}_{ij}\left(t\right),\theta_{ij}\left(t\right)\right]-I\gamma_{r}{\vec{\omega }}_{i}\left(t\right)+{\vec{Q}}_{i}\left(t\right)\;\;\equiv {\vec{\tau }}_{i}\left(t\right), \end{array}$$
where *I* is the moment of inertia of a particle, ${\vec{\omega }}_{i}$ is the angular velocity, $\gamma_{r}$ is the rotational viscous coefficient, and ${\vec{T}}_{i}$ is the magnetic torque (see Appendix B), which depends on the separations ${\vec{x}}_{ij}$ and angles $\theta_{ij}$ between all other *j* particles. ${\vec{Q}}_{i}$ is a random torque with statistics related to the temperature and viscous coefficient according to
(7)$$\begin{array}{ccc} \langle Q_{\alpha }\rangle & = & 0,\;\;\;\;\;\;\langle Q_{\alpha }\left(t\right)Q_{\beta }\left(t+t^{\prime }\right)\rangle =6m\gamma_{r}k_{B}T\delta_{\alpha \beta }\delta \left(t^{\prime }\right). \end{array}$$

The rotational viscous coefficient for spherical particles is given by $I\gamma_{r}=8\pi \eta R^{3}$, where η is the fluid viscosity and *R* is the hydrodynamic radius. Note that the ratio $\gamma_{r}/\gamma_{t}=10/3$ for perfect spheres [28]. The right-hand-side of Equation (Equation 6) is called $\tau_{i}$ as shorthand notation.

Once again, we assume that the particles that form agglomerates neither rotate truly independently, nor do they form rigid clumps that cannot change shape over time. To model ligand sticking, we therefore replace the rotational Equation (Equation 6) with the weighted average
(8)$$\frac{d{\vec{\omega }}_{i}}{dt}=\left(1-c\right)\frac{{\vec{\tau }}_{i}}{I}+\frac{c}{N_{agg}}\sum_{k=1}^{N_{agg}}\frac{{\vec{\tau }}_{k}}{{\vec{I}}_{agg}},$$
where *c* is the same phenomenological sticking parameter that was used in the corresponding translational Equation (Equation 5) above. In actual fact, these two parameters can be different. For example, the potential introduced by Wang et al., to account for ligand interdigitation/entanglement in simulations [3] resisted sliding of particles over one another, but there was no restriction on particles moving apart radially. For simplicity, we here take the rotational and translation sticking parameters to be the same. Additionally, note that components of the vector ${\vec{I}}_{agg}$ are the moments of inertia of the agglomerate around its center of mass, relative to rotation around the *x*, *y* and *z* axes.

The inter-particle interactions included are magnetic dipole–dipole interactions (repulsive and attractive) [29], van der Waals attraction and short-range repulsion [30,31]. The equations for the resulting forces are given in the Appendix B. Short-range repulsion prohibits particles from overlapping in space and provides an equilibrium separation for particles. Note that the magnetic dipole moments are assumed to be rigidly locked along a particular axis in the particle. In other words, magnetic relaxation only occurs through particle rotation (Brownian relaxation), and not through movement of the magnetic dipole moment within particles (Néel relaxation). We therefore only examine ferromagnetic particles with large anisotropy energy barriers (large anisotropy constant or large volumes) in this work. Few works have taken into account both mechanisms as the timescales are well-separated [32]. We also use a dipole cutoff shell of 10(r+L), outside which the dipole forces are neglected. As dipole forces decrease with separation as $D^{-4}$, the maximum forces at the cutoff shell are 10,000 times smaller than for particles with ligands in contact. However, we performed one simulation with the dipole cut-off extended to 20(r+L) and found many of our results did not change (for example, magnetization versus applied field loops) but that the shape of clumps did change. This will be discussed later. This one simulation took days longer to run than the others. In the future, our calculation will be extended to treat the dipole-dipole interactions using Ewald sums that reduce the computation time so that it scales as Nlog(N), rather than as N!, where *N* is the number of dipoles [33,34].

Equations of motion (Equation 5) and (Equation 8) are solved numerically using a modified velocity Verlet algorithm attributed to van Gunsteren and Berendsen [33,35] to solve the second-order, stochastic differential equations, with random forces and torques.

Magnetite nanoparticles are considered with magnetization $M=3.12\times 10^{5}$ A/m (corresponding to around 60 emu/g [15] since magnetite’s mass density is 5255 kg/m${}^{3}$), radius r=25 nm, ligand length L=5 nm, distance between ligand heads h=5 nm and temperature T=298 K. The hydrodynamic radius is assumed to be R=L+r=30 nm. The surrounding fluid is hexane with a viscosity of $\eta =2.97\times 10^{-4}$ N s/m${}^{2}$ and a corresponding Hamaker constant $A_{131}=29\times 10^{-21}$ J [36]. For assembly in a uniform field, 500 particles are simulated for 0.01 s of real time, with a time step of Δt=0.1 ns. Larger timesteps lead to an unphysical overlap of particles. The time step is less than the momentum relaxation time for the particles, hence the use of the van Gunsteren and Berendsen numerical integration scheme, rather than a Brownian scheme where the inertial term is neglected. Periodic boundary conditions were used along with a volume fraction of magnetic material corresponding to $7.58\times 10^{-4}$. This density corresponds to 4 mg/mL of magnetite in hexane, which is an appropriate concentration for applications. For all simulations the particles are started from random (although non-overlapping) positions and with random magnetization directions.

We ran our fortran code on 25 Intel(R) Xeon 2.2 GHz processors for 60–70 h to simulate 500 particles over 0.01 s of simulation time. Runs are repeated to ensure that the results are consistent.

For the hysteresis loops, 500 particles were again simulated. An applied field with frequency 1.25 kHz and amplitude 100 kA/m was used and simulations run through 10 cycles of the field. This is sufficient to see the loops reach a steady state.

### 2.2. Estimate of the Sticking Parameter

Here, we estimate the value of *c* for which the clumps formed in simulations transition from long chains to disordered agglomerates. We do this by comparing the time it takes a particle to diffuse ($t_{diff}$) towards another, to the time it takes it to rotate so that the magnetic dipolar interactions are minimized ($t_{rot}$). A similar comparison of diffusion and magnetic-relaxation timescales proved successful in predicting chaining in our recent work [37].

A characteristic rotation time for particles in a fluid experiencing a net torque is [32]
(9)$$t_{rot}∼\frac{8\pi \eta R^{3}}{\left(1-c\right)\tau_{total}},$$
where the total torque $\tau_{total}$ a particle in an agglomerate experiences is decreased due to stickiness by the term (1−c), consistent with Equation (Equation 6), and neglecting torques from far away particles (i.e., those not in the agglomerate). R∼(r+L) is the hydrodynamic radius of the particle. Equation (Equation 9) comes from the equation for the torque needed to keep a sphere rotating with uniform velocity in a still, infinite, viscous fluid [38].

This can be compared to the expected time for agglomeration based on the density of particles and the thermal energy. Let the time for a particle to diffusively travel a distance *x* to the region close to another particle be $t_{diff}$. It is estimated using the fluctuation dissipation theorem result as [39]
(10)$$t_{diff}=\frac{x^{2}6\pi \eta R}{k_{B}T}.$$

The distance *x* is the mean distance between the 500 particles initially in the simulation volume, i.e., $x={\left(V/N\right)}^{1/3}$, where *N* is the total number of particles and *V* is the total simulation volume. Substituting the parameters for 50 nm diameter magnetite particles into Equation (Equation 10) finds a diffusion time on the order of 1 ms.

When the time for a particle in an agglomerate to rotate so that its magnetic moment points in the local field direction is larger than the average time for another particle to join the agglomerate, we expect a messy, noncollinear agglomerate to form. The new particle may stabilize the noncollinear shape. One may imagine an incoming particle to have been affected by thermal translations and rotations as well as experiencing the complicated superposition of dipolar fields from all particles on its approach, which is why its magnetic moment is not automatically aligned with the field of the agglomerate it joins. The condition for messy agglomerate shapes is therefore $t_{diff}<t_{rot}$ (see Equations (Equation 9) and (Equation 10)).

Rearranging for *c*, one obtains
(11)$$c∼1-\frac{4k_{B}T}{3\tau_{total}}\frac{R^{2}}{x^{2}}.$$

Then, for a given particle concentration, temperature and particle size, the stickiness parameter required to achieve noncollinear agglomerate shapes can be estimated. Note that for larger thermal energies, less “stickiness” (lower ***c*** values) is required for messy clumps to form, as is to be expected. Furthermore, note here that the fluid viscosity is absent as both $t_{rot}$ and $t_{diff}$ are linearly dependent on viscosity and the term cancels.

Using the parameters for the 50 nm diameter magnetite nanoparticles and assuming a magnetic torque $\tau_{total}$ for two particles in contact with an angle of $5^{\circ }$ between their dipole moments, Equation (Equation 11) gives
(12)c>0.995
for messy agglomerates to form. This value can vary up to c>0.999 by assuming the angle between dipoles is $90^{\circ }$ before they begin to rotate to minimize the dipole–dipole energy. Therefore, in Langevin dynamics simulations at zero field, one expects to find chains of particle when c<0.995 and messy agglomerates for c>0.995. This matches exceptionally well with the simulation results, as will be shown below.

## 3. Results

### 3.1. Assembly in a Uniform, Constant Field

In Figure 1, a snap shot of the simulation volume is shown at the end of the 10 ms duration under different conditions. On the left-hand-side are the results when there is no applied field, for different values of the stickiness parameter *c* = (a) 0, (b) 0.9 and (c) 0.999. The trend downwards (increasing stickiness) is for messier clumps to form. As expected, when c=0 we see that agglomerates tend to be single particle width chains or loops with relatively small angles between the dipole moment of adjacent particles. When the sticking parameter is increased to c=0.999 the clumps form shapes other than chains and loops with significant bends. On the right-hand-side of Figure 1, the corresponding results are shown for an applied field of $\mu_{0}H=5$ mT in (d), (e) and (f). This field is large enough to eliminate any loops for c=0 (see panel (e)) and to considerably straighten the chains (it corresponds to an interaction energy $m\mu_{0}H∼0.39k_{B}T$). Some joining of single-width chains is seen. For strong ligand bridging/friction c=0.999 (bottom, right panel), shorter, more crooked chains form.

Of particular note is the fact that there appears to be a transition in behavior from c=0.9 (mainly straight lines in (e)) to c=0.999 (shorter, more disordered agglomerates in (f)). This is across our predicted critical value of c=0.995. The results must be explored quantitatively to support this point.

In order to numerically compare our simulation results to each other (and in the future to experimental results), we have used two characterization parameters. For the comparison of the clump shapes that form in a simulation, the average normalized length of clumps is calculated. This dimensionless parameter is calculated as the length along the long axis of a clump, divided by $2RN_{agg}$, where $N_{agg}$ is the number of particles in an agglomerate and *R* is the hydrodynamic radius. This gives a parameter equal to one for perfectly straight lines of connected particles. For spherical clumps of particles, the normalized length goes as $1/(N_{agg}^{2/3})$.

A dipolar order parameter is calculated as an accurate way of comparing the internal structure of clumps formed in various simulations. This is simply the total pairwise dipolar energy of the *N* particles in the simulation, divided by the dipolar energy of a perfect chain of *N* particles, with each dipole aligned tip-to-tail with its neighbor(s), namely
(13)$$Dipolar\;order=\frac{\sum_{i,j}\frac{1}{r_{ij}^{3}}\left(3\frac{\left(\vec{m_{i}}\cdot \vec{r_{ij}}\right)\left(\vec{m_{j}}\cdot \vec{r_{ij}}\right)}{r_{ij}^{2}}-\vec{m_{i}}\cdot \vec{m_{j}}\right)}{\sum_{i,j}\frac{2m^{2}}{r_{ij}^{3}}},$$
where $\vec{r_{ij}}$ is the vector between the centers of particles *i* and *j* and the sums are over all particle pairs in the simulation. This parameter accurately reflects the amount of dipolar ordering, regardless of the particular shape of clumps. In other words, the dipolar order of a perfectly straight chain or a perfect loop of particles that both have dipoles aligned tip-to-tail can be similar, even though their normalized length is very different.

Note that a fractal dimension may be used to characterize the shape of clumps of particles [6,40,41]. The results of our simulations over 0.01 s yield agglomerates that are too small (typically under 20 particles) for the fractal dimension to be a good characterization tool in this case.

In Figure 2, we show the normalized length (top panels) and dipolar order (bottom panels) for particles in no applied magnetic field (left) and a field of 5 mT (right), as a function of time. The characterization parameters tend to vary with time rapidly near the start of the simulation but settle into a nearly constant state by the end of the simulation. Corresponding with the shape of the clumps seen in Figure 1, for c=0 (blue dots) the dipolar order parameter is closer to 1 and the normalized length is also close to 1. As we increase the sticking parameter (red squares correspond to c=0.9 and green triangles to c=0.999) we see that agglomerates tend to be more disordered both in terms of their shape and their internal magnetic structure.

When the sticking parameter is increased to 0.999 (green triangles), which is above the critical value estimated above, one sees a sharp decrease in both the normalized length and the dipolar order. The simulations and analytic estimate are therefore in excellent agreement. Notice that the drop in normalized length is not so marked when an external field is applied (right panels in Figure 2) because the interaction with the applied field creates more ordered clumps.

Notice that Equation (Equation 11) for the critical sticking parameter does not have any dependence on the fluid viscosity. This means that the result is unchanged depending on the medium in which particles are suspended. What is important is the size of the magnetic torque through $\tau_{total}$, the temperature, the concentration of particles through *x*, and their hydrodynamic radius *R*. The larger the hydrodynamic radius, for example, the smaller the ligand sticking parameter can be to still induce disordered clumps of particles. However, note that this simple estimate does not take into account hydrodynamic interactions between particles and these interactions are expected to be appreciable when particles are near one another [23,24] and the fluid viscosity would indeed play a role.

In the presence of an applied field, the trend in the ordering parameter as a function of *c* is nonmonotonic. Here the normalized length is actually a little larger for c=0.9 (red squares) than for c=0 (blue dots), although both are close to 1 in Figure 2b. This slight increase can be explained by a damping of the random thermal motion as *c* increases and as particles clump together to form larger masses. This has a resulting increase in the linearity of the chains. However, when the sticking parameter is increased further, to c=0.999 (green triangles), as in the case of no applied field, we again see a decrease in the magnitude of both the dipolar order and the normalized length.

In the study of colloids, one is often interested in the initial rate of agglomeration and comparing that rate between different colloidal systems using the so-called “stability ratio” [42,43]. We can do that here by comparing the initial rate (t<0.1 ms) of agglomeration between systems with no sticking of ligands (c=0) to those with our phenomenological sticking turned on (c≠0). The case of zero applied field is considered to this purpose. We find that for c<0.999, there is little difference in the agglomeration rate between systems. This is supported by examining Figure 2c, where one sees that the blue and red data points lie almost on top of each other for the dipolar order as a function of time. Physically, this is consistent with the idea that particles only experience friction after they have already been drawn into an agglomerate, so the agglomeration rate should not markedly change when c≠0. However, this changes for a large sticking parameter c=0.999, and the rate of agglomeration slows down slightly then (again, see Figure 2c). This slow down is because the dipolar order in newly formed agglomerates is less, meaning that there is a weaker attractive magnetic force drawing in new particles to the agglomerate. Messy clumps have a lower net moment and so grow at a slower rate than ordered chains.

The agglomeration rate also depends on particle concentration, although only one concentration is considered throughout most of this article since the focus is on the effect of the sticking parameter *c*. We ran one simulation at half concentration (volume fraction of magnetic material is $3.79\times 10^{-4}$) and found that the initial agglomeration rate was roughly half of that seen in the other simulations.

### 3.2. Hysteresis Loops

The simulations are run for the same particle systems, but in an oscillating magnetic field appropriate for magnetic hyperthermia treatments. Namely, the frequency of the sinusoidal field is f=1.25 kHz and the amplitude is $H_{\max}=100$ kA/m (ensuring that $H_{\max}f<5\times 10^{8}$ Hz·A/m, as required for treatments [44]). Movies of the particle behavior in the fluid are provided in the Appendix A for c=0 (no ligand friction, individual particle behavior), c=0.9 and 0.99 (moderate friction), and for c=0.999 (strong friction between ligands, collective behavior of clumps). One finds that the magnetization reversal process looks very different in these cases. The particles that do not experience sticking/friction are able to rotate relatively freely in the oscillating field. On the other hand, the particles with significant sticking assemble into more rigid clumps that rotate more slowly—due to a larger moment of inertia—to follow the oscillating field. In fact, for c=0.999 (beyond the critical value), the particles cannot reverse with the field once clumps are formed, as their moment of inertia is too large and also because the assembled clumps are unable to bend or twist in order to assist their reorientation. To quantify this change in behavior with increasing *c*, we plot the net magnetization as a function of field in Figure 3.

Imaging the dynamics of nanoparticles in fluids may be difficult in real time, but the magnetization as a function of time for a sample can be measured relatively easily. In Figure 3, we present the normalized magnetization along the applied field axis, as a function of the applied field with frequency f=125 kHz. The case for c= (a) 0, (b) 0.9, (c) 0.99 and (d) 0.999 are drawn in the top four panels. The changing color of the points is designed so that one can follow the changing magnetization as a function of time, starting from red and going to red after five field cycles. The particles are started in random orientations at t=0 and so the initial magnetization is zero. Initially the field is positive, with the system developing a positive magnetic moment.

For c=0 in Figure 3a, a regular hysteresis loop develops quickly. Over five cycles of the field, the loop does not change markedly. There is a coercive field of roughly 0.02 T due to the finite time it takes for the particles to rotate to align with the applied field. Chains of particles form, but the individual particles still rotate almost independently to align with the applied oscillating field.

For c=0.9 in Figure 3b, the coercivity is larger than for c=0 as particles in chains move as single entities and therefore rotate to follow the field more slowly. In the first few cycles of the field, a shoulder is visible in the top left quadrant of the plot. This is a signature that first the single-particles rotate, and then the clumps following later. This difference in timescale is clearly seen in the video (see Appendix A). The magnetization shoulder is still visible after five field cycles, but disappears after 10 cycles as single particles are absorbed into agglomerates. From that point the hysteresis loop is in a steady state (see panel (e) for c=0.9). Note that for a larger fluid viscosity, it takes much longer for the steady state to be reached, but a very similar steady state was found to occur. However, we again point out that hydrodynamic interactions between particles are not accounted for in these simulations.

For c=0.99 in Figure 3c, the coercive field becomes very large (almost 0.125 T) as agglomerates become more rigid and also messier, increasing their reversal times. Again a “shoulder” is seen in the loops, that disappears on consecutive field cycling. The area enclosed by the M−H loop is much larger than in the previous two cases.

This trend of increasing M−H area does not continue, however. The case of c=0.999 is drawn in Figure 3d. The particle clumps that form are so rigidly-locked together and aligned with the initial positive field that they are unable to follow the applied field direction. In the corresponding video (see Appendix A), one can see the single particles flip up and down with the field, but the chains that form cannot flip. This gives rise to a hysteresis loop that has positive magnetization at all times. The M−H area is decreased compared to the other three cases.

In Figure 3e, hysteresis loops are drawn corresponding to (a–d) for one field cycle, after several cycles have occurred and the system has approached steady state. One sees that the hysteresis area increases and then decreases above the critical value of *c* (0.999). This has implications on magnetic hyperthermia. The area enclosed by global hysteresis loops—but not local ones [45]—is correlated with the amount of heat that can be generated. One sees that this heat can be increased or decreased by large amounts when ligand friction/stickiness is increased, depending on whether particle agglomerates have sufficient time to follow the oscillating applied field.

In Figure 4, the effect of increasing the dipole cut-off distance is explored. In panel (a), the initial hysteresis loops (starting from randomly placed particles) for c=0.9 are shown with a dipole cut-off of 10(r+L) (as seen before, color points) and 20(r+L) (black line). One sees that there are small differences, but the M−H loops settle to a similar shape and the coercive field is not affected. In panel (b), the particle positions at the end of the simulation are shown (magnetization near negative saturation) for dipole cut-off distance 10(r+L) (red particles) and 20(r+L) (blue particles). The results are overlaid to make a better comparison. There are some longer chains that form with the larger dipole cut-off, which is as to be expected. However, note that both cases show interesting clump shapes, beyond the single-width chains that are predicted by simulations without frictional forces included. This is of particular note as the magnetization is nearly saturated, yet some messy clumps remain.

## 4. Discussion

A particle-level Langevin dynamics simulation is presented with a phenomenological sticking parameter *c* included to model ligand interdigitation and ligand bridging in a computationally cheap manner. The ligand interdigitation and bridging restricts particles from translating or rotating independently when their ligand coatings are in contact. We see in a uniform, weak applied field that by increasing the value of the sticking parameter, the formed agglomerates’ shape transitions from linear, organized clumps to more disordered, complex clump shapes. We find through simulations that there is a critical sticking parameter for the transition between these two different behaviors, around c=0.999 for the 50 nm diameter magnetite particles used here. An analytic estimate agrees with this result. A phenomenological sticking parameter of c=0.999 corresponds to ligand forces on the order of 10 pN, on the same order as the deterministic forces in the system. This does not seem unreasonable given experiments on the binding of ligands to surfaces [26,27], which show using Atomic Force Microscopy that rupture forces for ligand-ligand or ligand-metal binding are on the order of 100–1000 pN.

In experiments, it is rare to see the self-assembly of long, straight chains of ferromagnetic particles (see for example Ref. [15]). Typically the clumps have a more disordered shape. Here the effect of ligand bridging and interdigitation is shown to help stabilize such shapes, that do not minimize the magnetic dipolar energy, whereas clumps that are highly ordered are simulated when ligand sticking is ignored. These calculations therefore provide a useful step to explaining such experiments. However, other explanations for the disordered clump shapes may include the fact that single particles are not perfectly spherical (thus the point dipole approximation breaks down) or that there are size distributions among particles [46,47]. The size of the magnetic moment and the density of particles may play a role in the shape of formed agglomerates [48,49]. Furthermore, taking into account hydrodynamic interactions between particles will change how they approach each other when self-assembling. All these effects therefore need to be considered in future work to provide a realistic model of self-assembly, in order to guide the design of materials for many applications.

We also present simulations for these magnetic particle systems in an oscillating field (f=1.25 kHz, $H_{max}=100$ kA/m) that is appropriate for hyperthermia treatments. The change in the particle dynamics is dramatic changing *c* from 0 to 0.999. Correspondingly, the change in the M−H area and the heat that may be produced is dramatic. At first the M−H area increases due to particle agglomerates rotating more slowly to align with the field, and then the area decreases when particle chains become locked in position. The effect of ligand bridging or interdigitation on hyperthermia could be tested in an experiment by preparing magnetic particle samples that have different ligand coatings (although the same length to keep dipole–dipole interaction strength constant) and comparing their specific absorption rates.

Only one applied field frequency and amplitude is presented here, but there is a whole range of particle parameters that may be explored in order to optimize heating. This work is exploratory in nature, and the effects of size, field amplitude, frequency and magnetization are yet to be explored in detail. As one example, we found that a higher frequency field changes the hysteresis loops presented in Figure 3 significantly as more agglomerates have a rotation time that is longer than the period of the field, causing more agglomerates to be locked in place.

It must be emphasized again that the magnetic moment is rigidly locked to the ferromagnetic particles in our simulations (Brownian rotations are considered, but not internal Néel rotations). For smaller particles—which are more realistic for biomedical applications—the Néel relaxation of the magnetization may be the dominant relaxation mechanism, even in agglomerates. Indeed, an “avalanche” of magnetic moment Néel reversals in chains of superparamagnetic particles was recently reported as giving rise to wide, square hysteresis loops [50]. These loops in turn enabled a greatly enhanced magnetic particle imaging (MPI) signal. An extension to our work would be to incorporate Néel relaxation mechanisms too, in order to tailor magnetic particle agglomerates for specific applications.

## Figures and Tables

**Figure 1 nanomaterials-11-02870-f001:**
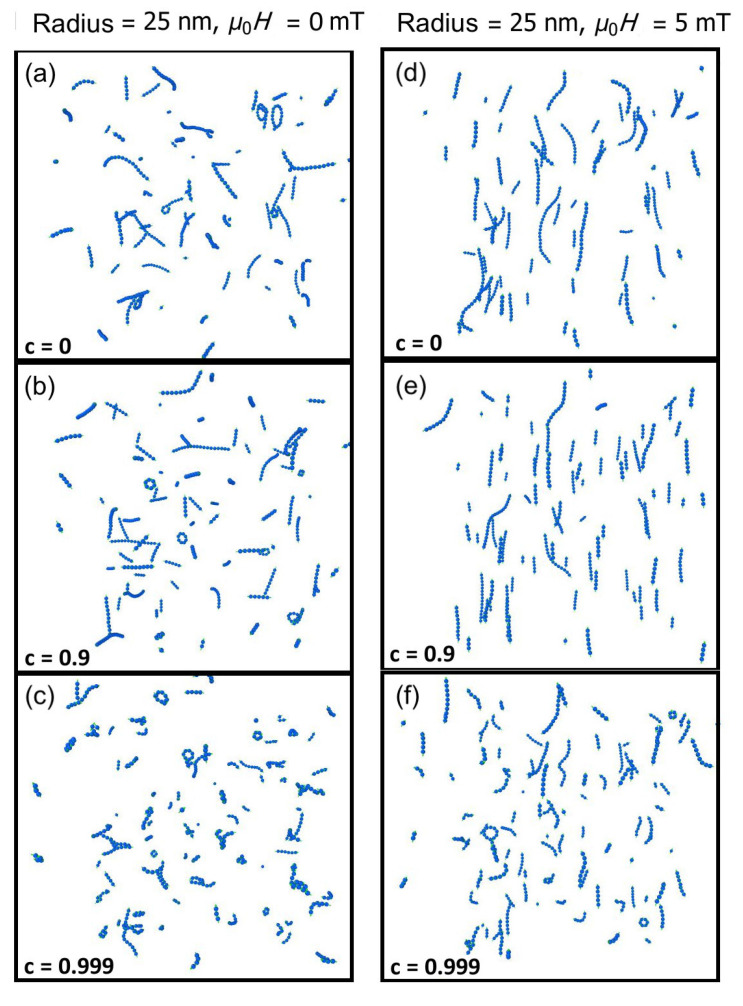
Characteristic simulation results in no field (left panels (**a**–**c**)) and a field of $\mu_{0}H_{0}=5$ mT (right panels (**d**–**f**)). The disorder in clumps tends to increase as the sticking parameter increases (higher sticking parameters in low panels).

**Figure 2 nanomaterials-11-02870-f002:**
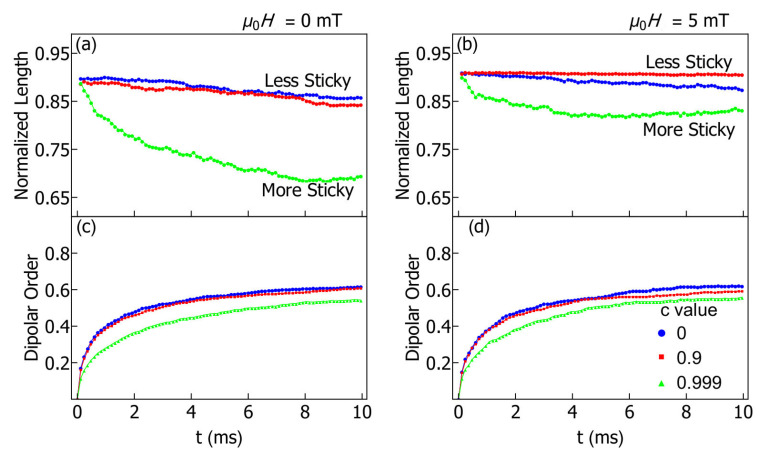
Characterization parameters for particles in no field (**a**,**c**) and a field of 5 mT (**b**,**d**). The top panels show the normalized length as a function of time (values closer to 1 indicate more chains) and the bottom panels show the dipolar order as a function of time.

**Figure 3 nanomaterials-11-02870-f003:**
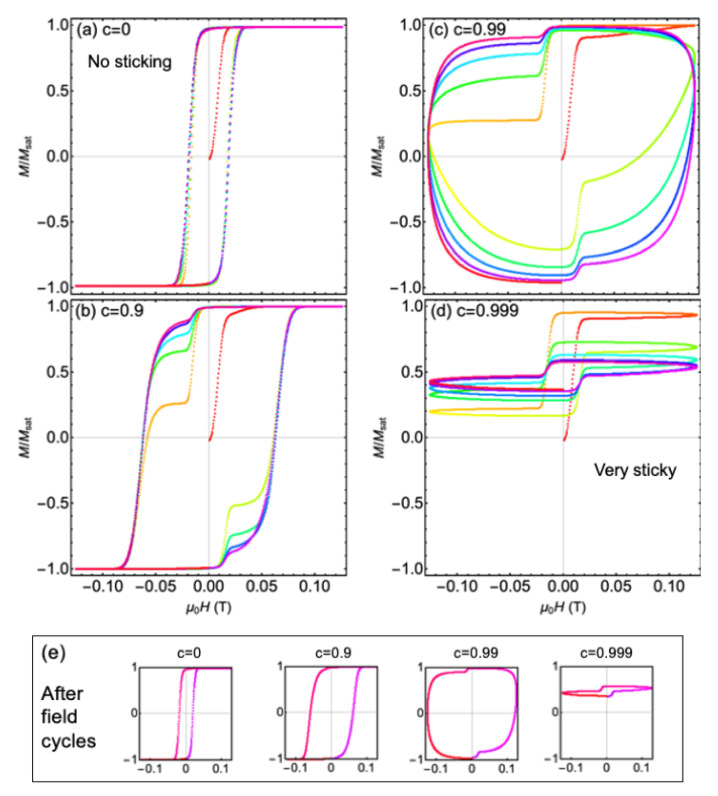
Magnetization versus applied field (f=1.25 kHz and $H_{max}=100$ kA/m) for different clumping constants: *c* = (**a**) 0, (**b**) 0.9, (**c**) 0.99 and (**d**) 0.999. In each of the four top panels, the particles are started at random positions and in zero field. As the field is applied, small clumps form, influencing the net magnetization in the oscillating field. The changing color of the points is designed so that one can follow the changing magnetization as a function of time, starting from red and going to red after five field cycles. In panel (**e**), the final hysteresis loops are shown in the four cases after 5 field cycles, when the loops are at a steady state. (For c=0.9, the loop in (**e**) is after 10 cycles rather than 5.)

**Figure 4 nanomaterials-11-02870-f004:**
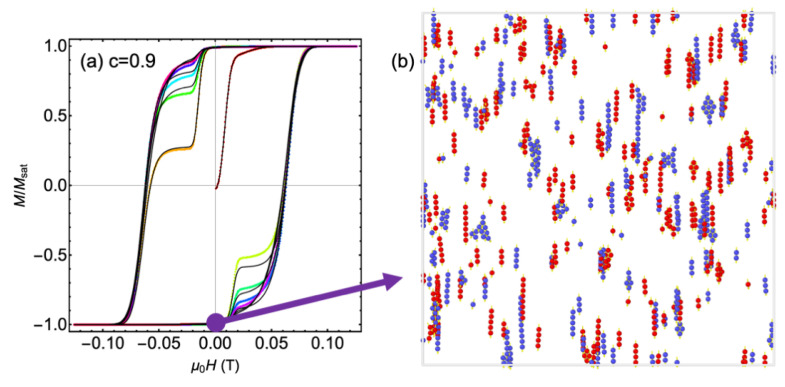
(**a**) The initial M−H loops for c=0.9 redrawn from Figure 3b (color) along with the same results using a larger dipole cut-off distance (black lines). After a few cycles of the field, there is little difference. (**b**) The configuration of magnetic nanoparticles at the end of both simulations, overlaid on top of one another. The red particles are with the smaller dipole cut-off of 10(r+L) and the blue particles are with the larger dipole cut-off of 20(r+L).

## Data Availability

The data presented in this study are available in the article and also inside the Appendix A. The Fortran code used for simulations is included as a Appendix A.

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
