# Peer review of "Simulating the Self-Assembly and Hysteresis Loops of Ferromagnetic Nanoparticles with Sticking of Ligands"

_nanomaterials, 2021, doi:10.3390/nano11112870_

Round 1
Reviewer 1 Report
This work reports on the results of numerical simulations of the impact magnetic nanoparticle self-sticking on particle clustering in a fluid and the effects of this clustering on dynamic hysteresis loops. Dynamics and various forces are incorporated within a Langevin description of linear and rotational particle motion that includes dipolar and other interactions, with the addition of a phenomenological sticking parameter. The advantage of this model compared to previous work of a similar nature is in its simplicity and computational speed. Notably, the model does not include Neel relaxation of the particle magnetization as well as possibly important hydrodynamic forces..
This work deals with a problem of current interest and the results should be useful for others working in this field. The manuscript is written in a clear and logical manner and deserves publication, after the following points are considered.
-
The model is new and includes many approximations and results are presented for only a very limited range of parameters. The results might best be described as preliminary or exploratory and perhaps the authors might consider this in the wording of the abstract and Introduction and Discussion.
-
Periodic boundary conditions are applied with a cut-off radius included for the long-range dipole interactions, rather than the typical treatment using the well-tested Ewald approximation. It would thus be more re-assuring if the impact of the value chosen for the cut-off radius were explored and mentioned, maybe on the loops.
-
It is explained that Eq. 11 has no dependence on viscosity. Would one expect viscosity to impact the MH loops?
-
In the Discussion, there appears “A phenomenological sticking parameter of c = 0.999 corresponds to ligand forces on the order of 10 pN, on the same order as the deterministic forces in the system. This does not seem unreasonable given experiments on the binding of ligands to surfaces.[26,27]” It would be useful to provide a more detailed explanation, of exactly how the experiments support this conclusion (what numbers are reported in the experiment?).
Author Response
Thank you to the reviewer for their comments.
Please see the attachment for our responses.

Reviewer 2 Report
Agglomeration of magnetic nanoparticles is an important subject now. This paper reports the dynamic behavior leading to the agglomeration of the particles in the absence of magnetic field and in the presence of 5 mT magnetic field by introducing the sticking parameter. The results will be useful for the experimental researchers engaging colloidal chemistry. Before accepting this paper for the publication, the following points should be considered or revised.
- The concept of sticking parameter is not clear. More understandable description or values of example is required.
- Figure 1 is the just result of the simulation. The distribution of the particles before the start of the reaction should be shown.
- In the experiment of agglomeration, the stability ratio (the ratio of the rate constants in two different conditions) is discussed. Is it possible to show any relationship to the stability ratio from the present simulation?
- Figure 2 show the dynamic results. Please discuss about the rate constant of the agglomeration from the results. Also, show the relationship between the rate constant and the initial particle concentration.
- Other small things; line 11, "nonmonotonic". line 123, "10(r + L)"
Author Response

(The authors gave the same response as above.)
